# Polyphenol-Derived Microbiota Metabolites and Cardiovascular Health: A Concise Review of Human Studies

**DOI:** 10.3390/antiox13121552

**Published:** 2024-12-18

**Authors:** Ana Clara da C. Pinaffi-Langley, Stefano Tarantini, Norman G. Hord, Andriy Yabluchanskiy

**Affiliations:** 1Oklahoma Center for Geroscience and Healthy Brain Aging, University of Oklahoma Health Sciences, Oklahoma City, OK 73117, USA; 2Department of Nutritional Sciences, College of Allied Health, University of Oklahoma Health Sciences, Oklahoma City, OK 73117, USA; 3Vascular Cognitive Impairment and Neurodegeneration Program, Department of Neurosurgery, University of Oklahoma Health Sciences, Oklahoma City, OK 73117, USA; 4International Training Program in Geroscience, Doctoral School of Basic and Translational Medicine, Department of Public Health, Semmelweis University, 1085 Budapest, Hungary; 5Department of Health Promotion Sciences, College of Public Health, University of Oklahoma Health Sciences, Oklahoma City, OK 73104, USA; 6Peggy and Charles Stephenson Cancer Center, University of Oklahoma Health Sciences, Oklahoma City, OK 73104, USA; 7Department of Nutritional Sciences, College of Education and Human Sciences, Oklahoma State University, Stillwater, OK 74075, USA

**Keywords:** polyphenols, gut microbiota, metabolites, cardiovascular health, urolithins, equols

## Abstract

Polyphenols, plant-derived secondary metabolites, play crucial roles in plant stress responses, growth regulation, and environmental interactions. In humans, polyphenols are associated with various health benefits, particularly in cardiometabolic health. Despite growing evidence of polyphenols’ health-promoting effects, their mechanisms remain poorly understood due to high interindividual variability in bioavailability and metabolism. Recent research highlights the bidirectional relationship between dietary polyphenols and the gut microbiota, which can influence polyphenol metabolism and, conversely, be modulated by polyphenol intake. In this concise review, we summarized recent advances in this area, with a special focus on isoflavones and ellagitannins and their corresponding metabotypes, and their effect on cardiovascular health. Human observational studies published in the past 10 years provide evidence for a consistent association of isoflavones and ellagitannins and their metabotypes with better cardiovascular risk factors. However, interventional studies with dietary polyphenols or isolated microbial metabolites indicate that the polyphenol–gut microbiota interrelationship is complex and not yet fully elucidated. Finally, we highlighted various pending research questions that will help identify effective targets for intervention with precision nutrition, thus maximizing individual responses to dietary and lifestyle interventions and improving human health.

## 1. Introduction

Polyphenols are plant secondary metabolites derived from the phenylpropanoid or polyketide pathways that carry more than one phenolic moiety [1]. Secondary metabolites are defined as organic molecules that are not essentially involved in the plant’s growth and reproduction. Instead, secondary metabolites are involved in processes such as biotic and abiotic stress response, animal and insect attraction or determent, and plant hormonal regulation [2,3]. In other words, polyphenols are part of the biological machinery that interfaces with the plant environment. Accordingly, environmental factors such as sun exposure, soil quality, and climate influence polyphenol biosynthesis and can lead to vastly different polyphenolic profiles in plants of the same species that were cultivated in different conditions [4]. This variability results in diverse chemical structures, from simple compounds like ellagic acid to complex polymeric compounds like proanthocyanidins.

In the context of human health, the association between consuming a diet rich in polyphenols and positive health outcomes is well established [5,6,7,8]. This is reflected in the recent expert guideline recommendations for flavan-3-ol–rich food intake for improving cardiometabolic health [9]. Despite these phenotypic observations, the underlying mechanisms through which polyphenols exert their physiological effects remain elusive, in large part due to the complexities of polyphenol bioaccessibility, -availability, and -activity. In recent years, the relationship between dietary polyphenols and the gut microbiota has emerged as a key factor in polyphenol and human health research. Dietary polyphenols and the gut microbiota have a bidirectional relationship, where they affect one another reciprocally—for instance, the gut microbiota can metabolize polyphenols into metabolites with greater bioavailability than their parent compounds and, at the same time, the presence of polyphenols in the intestinal lumen can favor the proliferation of certain bacterial species [10,11]. In fact, recent changes in the definition of prebiotics allowed for the inclusion of polyphenols in this classification [12].

Despite the advances in identifying potentially bioactive metabolites and in gut microbiota research, the main effectors of the two-way relationship between polyphenols and the gut microbiota and how this relationship can be targeted to improve human health remain uncertain. In this concise review, we aimed to summarize recent advances in this area, with a special focus on isoflavones and ellagitannins and their corresponding metabotypes, and their effect on cardiovascular health. To date, these are the only polyphenols with well-defined producer and non-producer metabotypes. We also aim to highlight future opportunities for research in this area to further evidence-based precision nutrition as a promising tool to effectively improve human health.

## 2. Dietary Polyphenols and the Gut Microbiota

Dietary polyphenols have poor bioavailability, meaning that the concentration of parent compounds (i.e., compounds originally found in foods) that enter peripheral circulation and can reach target tissues is low and kinetically disadvantaged. Instead, the small aglycones that can be absorbed into enterocytes via passive or active transport undergo phase I and phase II metabolism in the liver before reaching circulation. A small proportion of liver-metabolized polyphenols can enter the entero-hepatobiliary circulation, returning to the small intestine for further catabolism and/or excretion. Further, efflux transporters in enterocytes pump dietary polyphenols and their phase I/II metabolites back into the intestinal lumen for eventual excretion, effectively reducing their net absorption. Polymeric or conjugated polyphenols (i.e., polyphenols bound to large molecules such as polysaccharides) accumulate in the colon, where most polyphenol–microbial interactions take place. Microbial metabolites can be absorbed into colonocytes and undergo further biotransformation in the liver before reaching the bloodstream and target tissues, or they can be excreted in the feces. Circulating metabolites undergo kidney catabolism and are excreted in the urine. Figure 1 presents a simplified model for polyphenol absorption, metabolism, and excretion. As the digestive route of polyphenols is beyond the scope of this review, we direct the interested reader to available reviews on the topic [13,14,15,16].

Studies on polyphenol digestion and absorption estimate that over 50% of ingested polyphenols accumulate in the lower gastrointestinal tract, especially the colon, with this proportion increasing to over 90% for polymeric or conjugated polyphenols [17,18,19]. Resident bacteria can utilize accumulated polyphenols as a substrate, thereby generating smaller, more bioaccessible metabolites via several biotransformation reactions. These reactions are broadly categorized into (1) cleavage, (2) hydrolysis, and (3) reduction reactions [20,21]. (1) Cleavage reactions involve the breaking of covalent bonds, especially carbon–carbon and carbon–oxygen bonds. When these breaks occur in a lactone or phenolic ring, they cause ring opening. Bond cleavage can also occur in the side chains, leading to the removal of methyl (demethylation) or carboxyl (decarboxylation) groups. Bacterial strains that can catalyze cleavage reactions in polyphenols include those of the *Gordonibacter*, *Clostridium*, *Eubacterium*, *Eggerthella*, and *Slackia* species [22,23,24,25]. (2) Hydrolysis reactions involve the deconjugation of polyphenols from larger molecules such as sugars and organic acids. These aglycones can be absorbed through colonocytes or further catabolized by microbial populations. Hydrolysis reactions pertinent to polyphenol catabolism are de-glycosylation and ester hydrolysis. The de-glycosylation reaction involves the deconjugation of a saccharide moiety from the polyphenol via the breaking of glycosidic bonds, whereas the ester hydrolysis reaction involves the breaking of an ester bond. Bacterial strains that can catalyze hydrolysis reactions in polyphenols include those of the *Lactobacillus*, *Bifidobacterium*, *Bacteroides*, *Eubacterium*, and *Blautia* species [23,26,27,28]. (3) Organic reduction reactions involve hydrogen addition (hydrogenation) and loss of oxygen (de-hydroxylation). As polyphenolic compounds can have several hydroxyl groups, de-hydroxylation at different positions and to different degrees can generate a large variety of compounds. Bacterial strains that can catalyze reduction reactions in polyphenols include those of the *Gordonibacter*, *Lactonifactor*, *Eggerthella*, and *Eubacterium* species [22,29,30,31,32]. Most often, hydrolysis reactions seem to occur first, followed by cleavage and/or reduction reactions. Considering the vast number of dietary polyphenols identified to date [33], the number of possible microbial metabolites is equally large and an area of intensive active research.

The accumulation of polyphenols in the lower gastrointestinal tract also leads to changes in the gut microbiota composition, highlighting the reciprocal and complex interaction between dietary polyphenols and this quasi-organ. In fact, some authors describe polyphenols as prebiotics [34,35]. This assertion is supported by evidence in humans showing that dietary polyphenol intake is associated with a decrease in the population of harmful bacteria such as those of the *Enterobacteriaceae* family [36,37] and an increase in the population of commensal bacteria, such as those of the *Bifidobacterium* and *Lactobacillus* families [36,37,38,39,40]. However, the mechanisms through which polyphenols may modulate gut microbiota composition remain incompletely understood. Current evidence points to a combination of antimicrobial and prebiotic-like effects, where the decrease in pathogenic bacteria confers a survival and proliferation advantage to commensal bacteria, which is further promoted by the utilization of dietary polyphenols as substrates by probiotic bacteria [35].

## 3. Polyphenol-Derived Microbial Metabolites and Cardiovascular Health

Considerable research has focused on the effect of polyphenols on cardiovascular health outcomes since their recognition as bioactive compounds [41,42,43]. As in vitro and animal studies were translated into human studies, researchers encountered a large interindividual variability in response to interventions with dietary polyphenols [44,45]. As gut microbiota research advanced in parallel to polyphenol research, the reciprocal interaction between them became unveiled [46,47,48,49]. In this section, we will focus on the microbial metabolites associated with ellagitannins and isoflavones, the two polyphenol groups with the most well-established metabotypes and metabolites that have been investigated in human studies. Of note, recently, Iglesias-Aguirre et al. [50] reported two novel metabotypes associated with resveratrol intake: lunularin producers and non-producers. As this discovery is recent and the effects of these metabotypes on human health have not been explored yet, we did not include resveratrol and its associated microbial metabolites in this review.

For this review, we searched PubMed utilizing the search strings “(urolithins OR ellagic acid OR ellagitannins) AND (cardiovascular health OR cardiovascular disease)” and “(equols OR isoflavones) AND (cardiovascular health OR cardiovascular disease)” to identify studies. The retrieved studies were screened using the following inclusion criteria: (i) observational or interventional human studies, (ii) main outcome related to cardiovascular health, (iii) published in English, and (iv) published in the last 10 years.

### 3.1. Urolithins

Urolithins are microbial metabolites derived from ellagic acid and ellagitannins, which are a class of hydrolysable tannins in which ellagic acid moieties are esterified to a sugar molecule. These polyphenols are present in walnuts, pomegranates, and berries such as black raspberries and strawberries [33] (Table 1). Both ellagic acid and ellagitannins are poorly bioavailable and accumulate in the colon, where they are catabolized by resident microbes. A critical step in this catabolic process is the ester hydrolysis of ellagic acid moieties, thus making these molecules available for further catabolism. Cleavage of the ester bond in a lactone ring is the next step, leading to ring opening. This is followed by a decarboxylation reaction, thus generating the first urolithin metabolite, urolithin M5. This initial metabolite has five hydroxyl substituents, and subsequent sequential de-hydroxylation reactions generate tetra-, tri-, di-, and monohydroxy urolithin products [10,51] (Figure 2). These de-hydroxylation reactions can happen at different positions, thus generating a variety of urolithins, of which urolithins A and B are the most studied. Urolithins are, in turn, more bioavailable than their parent compounds. In a direct supplementation study [52], the oral intake of 500 mg of urolithin A led to plasma concentrations of phase II metabolite (urolithin A-glucuronide) as high as 481 ng/mL.

Ellagitannins intake is associated with well-defined metabotypes based on urinary excretion: metabotype 0, A, and B. Metabotype 0 is classified as a non-producer. In this metabotype, urolithin M5 is detected but dehydroxylase activity is not present or too low to efficiently convert urolithin M5 into other urolithins. Metabotypes A and B are classified as urolithin A and urolithin B producers, respectively. Metabotype B produces urolithin A, isourolithin A, and urolithin B, whereas metabotype A produces only urolithin A due the lack of 8-dehydroxylase activity [53]. Urolithin A is notable for being the first microbial metabolite to be commercially available as an isolated compound and dietary supplement. Safety studies in humans indicated that urolithin A is safe and well tolerated in the doses tested (250–2000 mg) [54], paving the way for the commercialization of other isolated metabolites.

Although the biological mechanisms through which urolithins exert their effects on cardiovascular health are incompletely understood, in vitro and in vivo studies indicate that these effects occur, at least in part, due to their capacity to modulate inflammatory signaling pathways in endothelial cells. Inflammation is accompanied by an increase in reactive oxygen species and oxidative stress, which promotes endothelial dysfunction, atherosclerotic processes, and thrombus formation [55]. Studies showed that treatment with urolithins inhibited myocardial fibrosis in an animal model of myocardial infarction [56] and protected against ischemia-reperfusion injury in animal models of cerebral [57] and myocardial [58] ischemia via activation of the Nrf2 signaling pathway, which upregulates the expression of antioxidant genes. Another animal study [59] demonstrated that urolithin A promoted atherosclerotic plaque stabilization via increased nitric oxide production and decreased YAP (Yes-associated protein)/TAZ (transcriptional coactivator with PDZ-binding motif) expression. YAP promotes TEAD transcriptional activity, which is associated with worsened endothelial inflammation and atherosclerotic plaque formation [60]. In human aortic endothelial cells, Spigoni et al. [61] demonstrated that urolithins increased nitric oxide production via increased activation of endothelial nitric oxide synthase, while Gimenez-Bastida et al. [62] reported that urolithin A protected these cells against TNF-α–induced inflammation and decreased the levels of C-C motif ligand 2, plasminogen activator inhibitor-1, and interleukin (IL)-8. Finally, Han et al. demonstrated, in two complementary in vitro studies [63,64], that urolithin exposure increased cholesterol efflux and decreased lipid accumulation in macrophages via modulation of the ERK pathway and a decrease in inflammatory cytokines like IL-6. Taken together, these results support the idea that urolithins can decrease inflammation and mitigate the oxidative damage that is associated with endothelial dysfunction and cardiovascular diseases. Table 2 presents the characteristics of human studies on urolithins and their precursors on cardiovascular health parameters published in the past 10 years.

To date, few human studies have been published on the effect of urolithins and/or their metabotypes on cardiovascular health. Interestingly, two studies [65,69] reported that obese and overweight individuals with metabotype B had worse lipid profile and cholesterol metabolism than those with metabotype A, while Gonzalez-Sarrias et al. [71] reported that only overweight or obese participants with metabotype B responded with an improvement in lipid profile to an intervention with ellagitannins. Further, Romo-Vaquero et al. [70] reported that individuals with metabotype B had an increased abundance of Coriobacteriaceae, which was in turn associated with increased cardiovascular risk. These results indicate that individuals with overweight or obesity, metabotype B, and dyslipidemia may benefit from dietary interventions with ellagitannins and that Coriobacteriaceae may mediate the worse lipid profile and cardiovascular risk observed in individuals with this metabotype. Istas et al. [68] reported significant acute improvements in flow-mediated dilation of the brachial artery, a gold-standard measurement of macrovascular endothelial function, in young men who consumed a red raspberry smoothie. Doubling the raspberry content of the smoothie did not result in further improvement in endothelial function, indicating a saturation or ceiling effect. In contrast, Nishimoto et al. [67] did not observe significant improvements in flow-mediated dilation of the brachial artery after direct supplementation with isolated urolithin A. Notably, this study only included participants who were classified as non-producers, whereas all participants in the study by Istas et al. [68] were classified as producers (80% metabotype A). Although providing the isolated metabolite may circumvent metabotype limitations, it is possible that other factors associated with the metabotype itself influence one’s ability to respond to the intervention.

### 3.2. Equols

Equols are microbial metabolites derived from isoflavones, mainly daidzein and genistein, which are a subclass of flavonoids. Isoflavones are found in soybean and soy-derived products such as soy milk, tofu, and miso, as well as green beans, chickpeas, and peanuts (Table 3) [33]. Although isoflavones are generally more bioavailable than other polyphenols [18,72], their absorption is still low, leading to their accumulation in the lower gastrointestinal tract and subsequent catabolism by resident bacteria. The first step in the catabolism of isoflavones is de-glycosylation, thus releasing aglycones from their sugar moiety. The aglycone (e.g., daidzein or genistein) first undergoes a hydrogenation reaction in the heterocycle and the product (dihydrodaidzein or dihydrogenistein) can be further catabolized in two distinct pathways: (1) the dihydroxy product can undergo a ring cleavage, leading to the production of *O*-desmethylangolensin, which may be further catabolized into small molecules such as phenol, resorcinol, and phloroglucinol; or (2) the dihydroxy product can undergo further reduction reactions in the heterocycle, namely hydrogenation and dihydroxylation reactions, to generate equols [73] (Figure 3). Similarly to urolithins, equols have greater bioavailability than isoflavones. In an early pharmacokinetic study [74], the plasma concentration of *S*-equol reached 1202 nmol/L after direct supplementation with a 20 mg bolus.

Isoflavone intake is associated with two distinct metabotypes: equol producers and non-producers. This classification is mainly based on urinary excretion after the intake of soy food products or isolated daidzein. As the name suggests, equol producers respond to isoflavone exposure with equol production, whereas non-producers respond to the same exposure with *O*-desmethylangolensin production instead [75]. Importantly, equol producers are most prevalent among populations with high intake of legumes like Asian and vegetarian populations, indicating a close relationship between metabotype and dietary habits [76,77].

Equols were the first polyphenol-derived microbial metabolites recognized for their biological effects independent of their parent compounds. Similarly to urolithins, the cardiovascular protective effects of equols are partially attributed to their antioxidant and anti-inflammatory activities. Further, due to the well-described estrogenic activities of equols [78,79], they have been widely studied in the context of menopause. Some of the early work focused on elucidating the mechanisms underlying the cardiovascular benefits of equols was conducted using human cell models. Different studies showed that exposure to equols increased nitric oxide production through endothelial redox signaling modulation [80], endothelial nitric oxide synthase activation [81], and decrease in superoxide levels and LDL cholesterol oxidation inhibition [82]. Munoz et al. [83] also reported that equols may decrease platelet activation and thrombus formation by an increased thromboxane A2 receptor antagonistic activity. In the context of menopause, studies with ovariectomized rats showed that equol supplementation increased nitric oxide production in the endothelium [84] and decreased superoxide levels and NADPH oxidase activity in the brain after cerebral ischemia. Interestingly, Zhang et al. [85] reported that estrogen receptor beta activation is involved in the antioxidant activity of equols via Nrf2 pathway modulation. More recently, Zhang et al. [86] demonstrated that equol supplementation decreased atherosclerotic lesions in APOE-deficient mice, and that this protective effect was mediated by an upregulation of the Nrf2 pathway. Taken together, these results indicate that equols are beneficial for cardiovascular health due, at least in part, to its antioxidant and anti-inflammatory effects, and that these effects may be particularly beneficial in menopause, when estrogen levels are depleted. Table 4 presents the characteristics of human studies on equols and their precursors on cardiovascular health parameters published in the past 10 years.

Isoflavone and equols are the most studied polyphenol–metabolite pair thus far, especially in the context of cardiovascular health and menopause symptom management. Recent studies exploring the association between equol or isoflavone levels or equol-producing status with cardiovascular disease risk factors [87,88,89,90,92,94,103] consistently reported lower indices of arterial stiffness, lower risk of hypertension, and better cardiometabolic profile in those classified as equol producers or with higher circulating equol or isoflavone levels. In contrast, three interventional studies [93,95,100] did not observe an improvement in lipid profile after supplementation with soy products, daidzein, or equol, even though one of the studies [93] reported an improvement in arterial stiffness after intervention with an equol supplement (98% S-equol). Interestingly, Hayashi et al. [91] reported that arterial compliance only improved in participants who were equol producers at baseline and assigned to the combined isoflavone supplementation and exercise intervention. This indicates that metabotype alone may be insufficient to benefit from an intervention with dietary polyphenols, and other lifestyle-related interventions can be combined to increase responsiveness, especially when considering that other factors such as exercise also influence gut microbiota status [104]. Similarly, Cavallini et al. [96] reported that cholesterol levels only improved in participants who received a fermented (probiotic) soy product with added isoflavones, indicating that interventions combining pre- and probiotics may be more effective, although equol status was not associated with this finding. 

Hazim et al. [97] reported that isoflavone supplementation induced an acute improvement in arterial stiffness in equol producers whereas equol supplementation did not acutely improve arterial stiffness in non-producers. This indicates that bypassing the metabotype with direct metabolite supplementation may not a straightforward solution and that there may be underexplored metabotype-associated health effects. However, the authors did not investigate whether direct equol supplementation could recapitulate the effects of isoflavone supplementation in equol producers. Some studies using serum levels of equols to classify participants as producers and non-producers used very different cutoff values (19 vs. 83 nM) [89,94], which obscures the true associations between equol-producing status and health outcomes. Setchell and Cole [76] proposed a log-transformed urinary equol-to-daidzein ratio to mitigate the interference of interindividual pharmacokinetics and analytical methodologies differences, although this approach has not been widely used.

## 4. Conclusions and Future Directions

The advances in polyphenol–gut microbiota interrelationship and human health research are progressing rapidly, in large part bolstered by technological advances in gut microbiota research (e.g., meta-omics techniques). As such, more metabotypes are being identified and described in the literature, such as the lunularin metabotype associated with resveratrol intake [50]. However, research on the clustering of different metabotypes is still emerging [105]. Understanding how metabotypes relate to one another and investigating possible networks will reveal additional inputs to modulate this system and contribute to advances in precision nutrition.

Another pending research question is how to unravel the purported health effects of polyphenol-derived microbial metabolites themselves from those of the microbial community. Are these metabolites simply biomarkers of a microbial community primed to catabolize polyphenols or are they affecting health changes independently (perhaps synergistically?) of the gut microbiota? There is some evidence for the independent effect of these microbial metabolites from studies that used isolated forms of specific metabolites [54,106], but an important caveat is that these studies utilize pharmacological doses of metabolites given orally and therefore the effect of physiological doses achieved with habitual food intake remains unclear.

Isolating bioactive microbial metabolites as dietary supplements is an alluring strategy for benefitting from their health effects, especially for those who are “non-producers”. However, there is little information on the effect of feeding high doses of an isolated metabolite on the gut microbiota profile or metabotype. Nishimoto et al. [67] described an increase in alpha diversity in non-producers after intervention with isolated urolithin A but no other changes. Moreover, the urolithin A concentration utilized in this study is one-fifth of the dose in commercially available dietary supplements. Further, as the gut microbiota is highly responsive to dietary inputs, “non-producers” may convert to “producers” (and vice versa); for instance, Gonzalez-Sarrias et al. [71] reported that half of those classified as non-producers at baseline became producers after a short intervention period with pomegranate extract. However, the rate of metabotype conversion and what type of dietary interventions may promote this conversion remain underexplored. 

Finally, Bae et al. [107] provided evidence of a critical mechanism by which gut bacterial enzymes diminish the beneficial effects of dietary polyphenols. This indicates that inter-individual variations in response to dietary polyphenols may be partially due to the presence of “antagonistic” gut bacteria as well as due to a lack of metabotype-associated bacterial species. As these and other research questions are probed, effective targets for intervention with precision nutrition will emerge, maximizing individual responses to dietary and lifestyle interventions and, ultimately, improving human health.

## Figures and Tables

**Figure 1 antioxidants-13-01552-f001:**
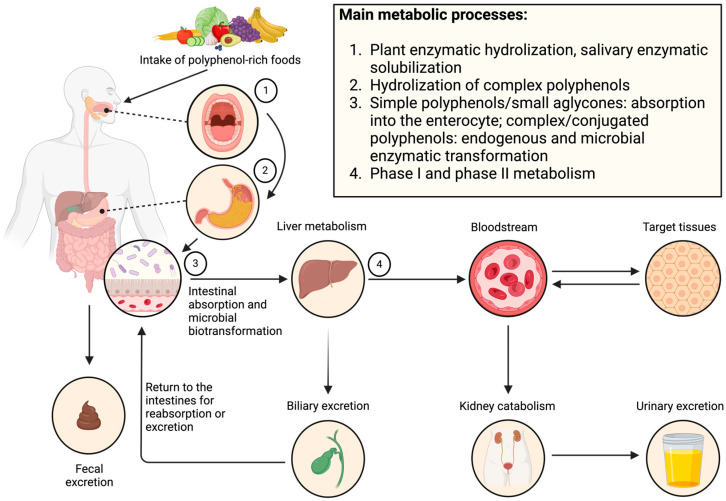
Simplified route of digestion, absorption, metabolism, and excretion of dietary polyphenols. Intact polyphenols are poorly absorbed in the gastrointestinal tract. Small aglycones are absorbed in the small intestine and undergo phase I and II liver metabolism before reaching the bloodstream and target tissues. Complex polyphenols accumulate in the colon, where they undergo extensive catabolism by the gut microbiota. Microbial metabolites can then be absorbed and undergo further biotransformation before reaching circulation and target tissues. Created with BioRender.com.

**Figure 2 antioxidants-13-01552-f002:**
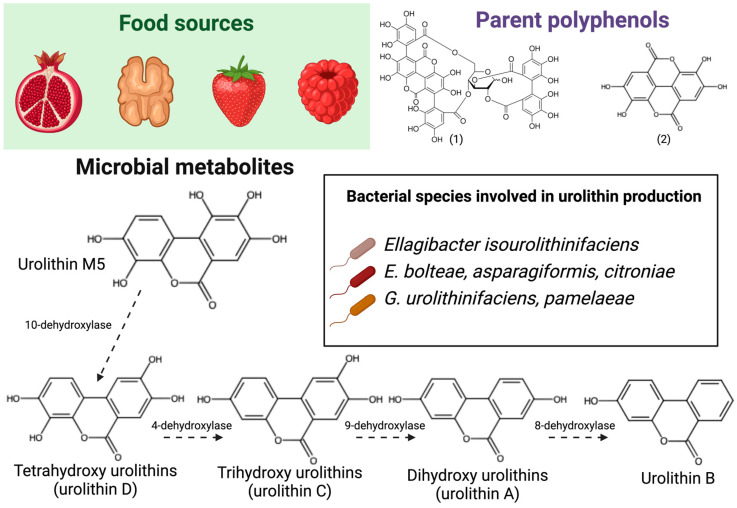
Pomegranates, walnuts, and berries like strawberries and raspberries are dietary sources of parent polyphenolic compounds ellagitannins, such as (1) β-punicalagins, and (2) ellagic acid. Upon ingestion, these polyphenolic compounds are poorly bioavailable and accumulate in the lower gastrointestinal tract. Bacterial species in the colon are involved in the catabolism of ellagitannins and ellagic acids. Main catabolic reactions are ester hydrolysis to release ellagic acid from ellagitannins, cleavage of the carbon–oxygen bond to open a lactone ring, and sequential de-hydroxylation reactions to generate urolithins with different degrees of hydroxylation. Among urolithins, urolithins A and B are the most studied in the context of human health. Created with BioRender.com. Chemical structures created with Marvin JS (Chemaxon, Budapest, Hungary).

**Figure 3 antioxidants-13-01552-f003:**
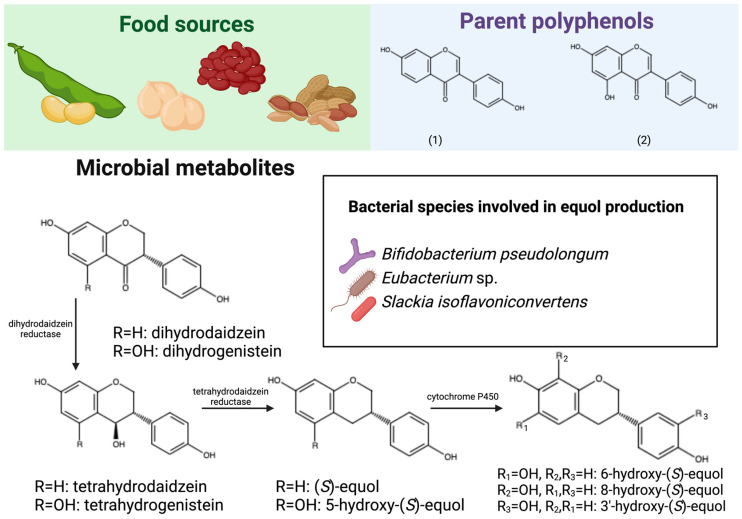
Legumes like soybeans, chickpeas, beans, and peanuts are dietary sources of parent polyphenolic compounds (1) daidzein and (2) genistein (isoflavones). Upon ingestion, these polyphenolic compounds are poorly bioavailable and accumulate in the lower gastrointestinal tract. Bacterial species in the colon are involved in the catabolism of isoflavones. Main catabolic reactions are de-glycosylation to release the polyphenolic aglycone and sequential hydrogenation and dehydroxylation reactions, generating equols. The alternative pathway involving ring cleavage and generation of *O*-desmethylangolensin is not shown in this figure. Created with BioRender.com. Chemical structures created with Marvin JS (Chemaxon, Budapest, Hungary).

**Table 1 antioxidants-13-01552-t001:** Food sources of ellagic acid and ellagitannins.

Polyphenolic Compound	Food Source (Mean Content ± SD) ^1^
Ellagic acid	Blackberry (44 ± 25 mg/100 g FW)
	Black raspberry (38 ± 0 mg/100 g FW)
	Pomegranate juice, pure (2.1 ± 1.5 mg/100 mL)
	Strawberry (1.2 ± 0.8 mg/100 g FW)
	Walnut (6 ± 2 mg/100 g FW)
Punicalagin	Pomegranate juice, pure (44 ± 71 mg/100 mL)
Sanguiin H-6	Raspberry (76 ± 0 mg/100 g FW)
Lambertianin C	Raspberry (31 ± 0 mg/100 g FW)

SD: standard deviation. ^1^ Source: phenol-explorer.eu.

**Table 2 antioxidants-13-01552-t002:** Human studies on urolithins and their parent polyphenols and cardiovascular health published in the past 10 years.

First Author (Year)	Study Design	Population	Intervention	Duration	Comparator	Primary Outcome	Main Finding
Cortes-Martin (2024) [65]	Prospective crossover interventional	49 adults (65% male) over 40 years of age	Oral ellagitannin supplementation (pomegranate extract capsules)	6 months	Urolithin metabotype status	Association between urolithin metabotype and change in bile acid and cholesterol metabolism after intervention	At baseline, urolithin-B producers had higher levels of fecal coprostanol and bile acids compared with other metabotypes (*p* < 0.05). Higher fecal urolithin A concentration was associated with lower levels of fecal coprostanol and bile acids (*p* < 0.05).
Laveriano-Santos (2022) [66]	Cross-sectional observational	560 adolescents (46% male) between 12 and 16 years old	N/A	N/A	Urine polyphenol-derived microbial metabolite levels	Association between microbial metabolites and metabolic syndrome features	Urolithin A was inversely associated with diastolic blood pressure (β: −0.02 [95% CI −0.03 to −0.01]). Urolithin B was inversely associated with metabolic syndrome score (β: −0.08 [95% CI −0.12 to −0.04]) and waist circumference (β: −0.03 [95% CI −0.04 to −0.01]). Higher urolithin B levels were associated with lower odds of abdominal obesity (OR: 0.94 [95% CI 0.89–0.98]) and lower odds of blood glucose levels above 110 mg/dL (OR: 0.92 [95% CI 0.88–0.96]).
Nishimoto (2022) [67]	Interventional	35 adults (69% male) between 40 and 65 years old	Urolithin A capsule (10 or 50 mg/day)	12 weeks	Placebo	Brachial artery flow-mediated dilation	For the 10 mg/day group, flow-mediated dilation showed a trend toward increase after 8 weeks of intervention (*p* = 0.078). Participants who responded to the intervention had lower flow-mediated dilation scores and low Bacillota–Bacteroidota ratio at baseline.
Istas (2018) [68]	Interventional	10 adults (100% male) between 18 and 35 years old	Red raspberry smoothie	24 h	Nutrient-matched control drink	Brachial artery flow-mediated dilation	Flow-mediated dilation increased acutely after consuming smoothie containing 200 g of raspberry compared with control drink (T + 2 h: 1.6% [95% CI 1.2–1.9%]; T + 24 h: 1% [95% CI 0.6–1.2%]; *p* < 0.0001). Increases were still significant but more modest when raspberry content was doubled.
Selma (2018) [69]	Cross-sectional observational	119 adults (60% male) over 18 years of age ^1^	N/A	N/A	Urolithin metabotype status	Association between urolithin metabotype and cardiometabolic risk factors	Overweight or obesity with urolithin-B metabotype was associated with worse lipid profile (*p* < 0.05 for total cholesterol, LDL-c, oxLDL-c, and apoB:apoA-I ratio). Overweight or obesity with urolithin-A metabotype was associated with a better lipid profile (*p* < 0.05 for apolipoprotein A-I and HDL-c).
Romo-Vaquero (2018) [70]	Interventional ^2^	249 adults (43% male) over 18 years of age	Oral supplementation with walnuts or pomegranate extract	3 days	Urolithin metabotype status	Association between specific urolithin metabotype and cardiovascular disease risk factors	Metabotype B was associated with increase abundance of Coriobacteriaceae, which was in turn positively associated with BMI, total cholesterol, LDL-c, and apoA (all *p* < 0.05).
Gonzalez-Sarrias (2017) [71]	Prospective crossover interventional	49 adults (65% male) over 40 years of age	Oral ellagitannin supplementation (pomegranate extract capsules)	6 months	Urolithin metabotype status	Association between urolithin metabotype and change in cardiovascular risk biomarkers after intervention	Only urolithin-B producers responded to the intervention with improvement in lipid profile, including reductions in total cholesterol (−16%), LDL-c (−15%), and non-HDL-c (−11%) (all *p* < 0.05).

^1^ Pooled baseline data from three interventional trials. ^2^ Pooled data from three interventional trials. CI: confidence interval, OR: odds ratio, LDL-c: LDL cholesterol, oxLDL-c: oxidized LDL cholesterol, apoA: apolipoprotein A-I, apoB: apolipoprotein B, HDL-c: HDL cholesterol, BMI: body mass index.

**Table 3 antioxidants-13-01552-t003:** Food sources of isoflavones.

Polyphenolic Compound	Food Source (Mean Content ± SD) ^1^
Daidzein	Edamame (0.6 ± 0.4 mg/100 g FW)
	Miso paste (4.9 ± 1.5 mg/100 g FW)
	Peanut (0.5 ± 0 mg/100 g FW)
	Soy milk (0.3 ± 0.4 mg/100 mL)
	Tofu (1.4 ± 2.5 mg/100 g FW)
Daidzin	Edamame (3.8 ± 16.4 mg/100 g FW)
	Miso paste (6.3 ± 4.6 mg/100 g FW)
	Soy milk (4.1 ± 8.1 mg/100 mL)
	Tofu (6.2 ± 8.4 mg/100 g FW)
Genistein	Black beans (0.6 ± 0 mg/100 g FW)
	Edamame (0.5 ± 0.6 mg/100 g FW)
	Miso paste (7.3 ± 7.3 mg/100 g FW)
Genistin	Edamame (4.6 ± 15.3 mg/100 g FW)
	Miso paste (10.8 ± 4.4 mg/100 g FW)
	Soy milk (4.9 ± 12.4 mg/100 mL)

SD: standard deviation. ^1^ Source: phenol-explorer.eu.

**Table 4 antioxidants-13-01552-t004:** Human studies on equols and their parent polyphenols and cardiovascular health published in the past 10 years.

First Author (Year)	Study Design	Population	Intervention	Duration	Comparator	Primary Outcome	Main Finding
Liang (2024) [87]	Prospective observational	305 adults (45% male) over 17 years of age	N/A	1 year	Urinary equol concentration tertiles	Association between urinary equols and cardiometabolic risk factors	Higher concentration of equols in urine was associated with decreased triglycerides levels, plasma atherogenic index, and metabolic syndrome score (*p* < 0.05).
Zhang (2022) [88]	Cross-sectional observational	313 adults (100% male) between 40 and 49 years old	N/A	N/A	Equol-producing status	Association between serum equols and aortic atherosclerosis	Equol producers (serum levels above 19 nM) had lower aortic calcification scores than non-producers (*p* > 0.05).
Zhang (2022) [89]	Cross-sectional observational	979 adults (100% male) between 40 and 79 years old	N/A	N/A	Equol-producing status tertile	Association between equol-producing status and aortic calcification	Equol producers had lower odds of having aortic calcification than non-producers (OR: 0.62 [95% CI 0.39–0.98]).
Zuo (2021) [90]	Prospective observational	2572 adults (30% male) between 40 and 75 years old	N/A	9 years	Serum and urinary isoflavone and equol concentrations tertile	Association between isoflavone and equol levels with increase in carotid intima-media thickness	Those in the highest tertile had smaller yearly changes in carotid intima-media thickness compared with those in the lowest tertile (*p*-interaction < 0.05).
Hayashi (2021) [91]	Prospective interventional	43 adults (100% female) between 45 and 69 years old	Aerobic exercise training plus oral isoflavone supplementation or oral isoflavone supplementation only	8 weeks	Equol producer status at baseline	Change in carotid arterial compliance	No changes in the supplementation-only group regardless of equol-producing status. In the supplementation plus exercise group, central arterial compliance increased only in the equol producers (*p* < 0.05).
Lee (2021) [92]	Nested case–control	229 adults with hypertension and 159 healthy controls over 35 years of age	N/A	N/A	Plasma isoflavone and equol concentrations tertile	Association between plasma isoflavones and equols with hypertension	Those in the highest equol level tertile had a lower risk of hypertension (OR: 0.34 [95% CI 0.20–0.57]).
Yoshikata (2018) [93]	Interventional	74 women in postmenopause between 44 and 74 years old	Oral equol supplementation	12 months	Cardiovascular parameters at baseline	Change in lipid profile and brachial-ankle pulse wave velocity as a measure of arterial stiffness	Equol supplementation increased HDL-c, LDL-c, and total cholesterol in postmenopausal women (all *p* < 0.01). Brachial-ankle pulse wave velocity was improved (*p* < 0.01).
Ahuja (2017) [94]	Cross-sectional observational	272 adults (100% male) between 40 and 49 years old	N/A	N/A	Equol-producer status	Association between equol-producer status and coronary artery calcification	Equol producers (serum levels above 83 nM) had lower odds of having coronary aortic calcification than non-equol producers (OR 0.09 [95% CI 0.01–0.89], *p* = 0.04).
Richter (2017) [95]	Prospective crossover interventional	20 adults (45% male) between 35 and 60 years old	Soy protein isolate powder	24 weeks	Isoflavone-free soy protein isolate powder	Change in ex vivo cholesterol efflux	No change in cholesterol efflux compared with control (*p* > 0.05).
Cavallini (2016) [96]	Prospective interventional	49 adults with hypercholesterolemia (100% male) between 37 and 57 years old	Probiotic soy product with or without isolated isoflavones supplementation	42 days	Non-fermented soy product	Change in lipid profile	Those who received the supplemented fermented soy product had a greater reduction in total cholesterol (−13.8 ± 7.7%) than those in the other two groups (*p* < 0.05). They also had a greater decrease in LDL-c and non-HDL-c than those in the comparator group only (*p* < 0.05). Equol producer status in the was inversely associated only with LDL(-) (*p* < 0.05).
Hazim (2016) [97]	Crossover interventional	28 adults (100% male) between 50 and 75 years of age	Oral isoflavone supplementation (producers and non-producers) or oral S-equol supplementation (non-producers only)	2–24 h	Placebo	Change in cfPWV, blood pressure, RHI, and nitric oxide	Among equol producers, isoflavone supplementation improved arterial stiffness after 24 h compared with placebo (*p* < 0.01). This improvement was associated with plasma equol levels (*p* = 0.01). Among non-producers, S-equol supplementation did not improve outcomes despite high plasma equol levels.
Reverri (2015) [98]	Crossover interventional	17 adults (29% male) over the age of 45 years	Soy nuts	4 weeks	Macronutrient-matched control snack	Change in inflammatory and oxidative biomarkers and endothelial function (RHI and AIx)	Intervention with soy nuts improved AIx (*p* = 0.03). Other outcomes were unchanged.
Acharjee (2015) [99]	Crossover interventional	60 women in postmenopause (average age: 55 years)	Therapeutic lifestyle intervention plus soy nuts	8 weeks	Therapeutic lifestyle intervention alone	Change in cardiovascular risk factors	Improvements in diastolic blood pressure (−7.7%, *p* = 0.02), C-reactive protein (−21.4%, *p* = 0.01), triglycerides (−22.9%, *p* = 0.02), and sICAM (−7.3%, *p* = 0.03) were only detected in equol producers with metabolic syndrome at baseline.
Liu (2015) [100]	Prospective interventional	270 women in postmenopause between 48 and 65 years old	Soy flour or isolated daidzein powder	6 months	Nutrient-matched control powder	Change in ambulatory blood pressure and flow-mediated dilation of the brachial artery	No differences among the three groups after the interventions.
Liu (2014) [101]	Prospective interventional	270 women in postmenopause between 48 and 65 years old	Soy flour or isolated daidzein powder	6 months	Nutrient-matched control powder	Change in cardiovascular risk biomarkers and carotid intima-media thickness	Supplementation with soy flour improved LDL-c (*p* = 0.006) and C-reactive protein (*p* = 0.022) levels compared with supplementation with isolated daidzein and control.
Qin (2014) [102]	Prospective interventional	210 adults with hypercholesterolemia (43% male) between 40 and 65 years old	Isolated daidzein	6 months	Placebo	Change in circulating cardiovascular risk biomarkers	Supplementation with daidzein decreased triglycerides and uric acid (all *p* < 0.05).
Shi (2014) [103]	Cross-sectional observational	299 pregnant women (average age: 28 years)	N/A	N/A	Urinary isoflavone concentration quartiles	Association between urinary isoflavone levels and cardiometabolic risk biomarkers	Higher urinary isoflavone concentrations were associated with lower fasting glucose, insulin, triglyceride, and IR-HOMA levels (all *p*-trend < 0.05).

OR: odds ratio, CI: confidence interval, LDL-c: LDL cholesterol, HDL-c: HDL cholesterol, LDL (-): electronegative LDL cholesterol, cfPWV: carotid-femoral pulse wave velocity, RHI: reactive hyperemia index, AIx: augmentation index, IR-HOMA: homeostatic model assessment of insulin resistance.

## Data Availability

No new data were created or analyzed in this study. Data sharing is not applicable to this article.

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
