# Peer review of "Polyphenol-Derived Microbiota Metabolites and Cardiovascular Health: A Concise Review of Human Studies"

_antioxidants, 2024, doi:10.3390/antiox13121552_

Round 1
Reviewer 1 Report
Well written manuscripts, covering all necessary aspects in easy tot read review article. Only one minor comment: r135/136 Lactobacilli are not obligate anaerobes but facultative anaerobes, just as most Enterobacteriaceae. The overall message is however correct, namely obligate anaerobes are often more on the healthy side
n.a.
Author Response
Major Comment:
Well written manuscripts, covering all necessary aspects in easy tot read review article. Only one minor comment: r135/136 Lactobacilli are not obligate anaerobes but facultative anaerobes, just as most Enterobacteriaceae. The overall message is however correct, namely obligate anaerobes are often more on the healthy side
We thank the reviewer for taking the time to review our manuscript and for their relevant comment. To avoid any misinterpretation or generalization, we have rewritten the sentence as "This assertion is supported by evidence in humans showing that dietary polyphenol intake is associated with a decrease in the population of harmful bacteria such as those of the Enterobacteriaceae family [36,37] and an increase in the population of commensal bacteria, such as those of the Bifidobacterium and Lactobacillus families [36-40]."
Reviewer 2 Report
This is a well-written review, which provided an excellent summary and valuable insights on polyphenol-derived microbiota metabolites and its impact on cardiovascular health. I have no objection to its publication in Antioxidants.
I have no comments.
Author Response
Major comments
This is a well-written review, which provided an excellent summary and valuable insights on polyphenol-derived microbiota metabolites and its impact on cardiovascular health. I have no objection to its publication in Antioxidants.
We thank the reviewer for taking the time to review our manuscript and for their careful consideration.
Reviewer 3 Report
The article evaluates and interprets research results related to the intervention of polyphenol-derived microbiota metabolites in cardiovascular health, highlighting a series of deficiencies in research conducted in the last ten years in the field of interactions between some dietary polyphenols (isoflavones and ellagitannins) and gut microbiota.
The article is well organized, systematized, the results of research conducted in the last ten years are analyzed with the highlighting of inadequate premises that lead to inconclusive results that leave the addressed issue not completely elucidated. The authors draw attention to these aspects and thus orient future research towards resolving these shortcomings.
Several aspects related to the content of the manuscript should be clarified:
The authors analyze only data related to some polyphenolic compounds - ellagitannins and isoflavones. Why was the range of polyphenols addressed restricted?
In the Conclusions there is information related to resveratrol - a compound that is not addressed in the other chapters. The information about of the resveratrol should also be included in the introduction or discussions not only in the conclusion, if they serve to argue some observations related to the analyzed polyphenolic compounds.
A synthetic presentation in a table of food sources of ellagitannins and isoflavones with their content specification and with the enumeration of the most important structures of these structural categories could be added in the chapter 2.
All of this could ensure a better understanding of the text and an improvement in the value of the manuscript.
Author Response
Major comments
The article evaluates and interprets research results related to the intervention of polyphenol-derived microbiota metabolites in cardiovascular health, highlighting a series of deficiencies in research conducted in the last ten years in the field of interactions between some dietary polyphenols (isoflavones and ellagitannins) and gut microbiota.
The article is well organized, systematized, the results of research conducted in the last ten years are analyzed with the highlighting of inadequate premises that lead to inconclusive results that leave the addressed issue not completely elucidated. The authors draw attention to these aspects and thus orient future research towards resolving these shortcomings.
We thank the reviewer for taking the time to review our manuscript. Please find point-by-point responses below and the corresponding revisions marked with track changes in the revised manuscript file.
Several aspects related to the content of the manuscript should be clarified:
The authors analyze only data related to some polyphenolic compounds - ellagitannins and isoflavones. Why was the range of polyphenols addressed restricted?
We thank the reviewer for this comment. Although all polyphenolic compounds have associated microbial metabolites, only isoflavones and ellagitannins have been associated with well-defined producer and non-producer metabotypes (equol producer and non-producer for isoflavones; metabotypes U-A, U-B, and U-0 for ellagitannins). Further, other polyphenol-derived microbial metabolites have an insufficient number of human studies for a review. This is why we chose to limit the range of possible polyphenols to ellagitannins and isoflavones. This is specified in the text in lines 70-71 and 148-151.
In the Conclusions there is information related to resveratrol - a compound that is not addressed in the other chapters. The information about of the resveratrol should also be included in the introduction or discussions not only in the conclusion, if they serve to argue some observations related to the analyzed polyphenolic compounds.
We thank the reviewer for this suggestion. We included information related to lunularin metabotypes and resveratrol earlier in the manuscript (lines 151-155), as follows:
"Of note, recently, Iglesias-Aguirre et al. [50] reported two novel metabotypes associated with resveratrol intake, lunularin producers and non-producers. As this discovery is recent and the effects of these metabotypes on human health have not been explored yet, we did not include resveratrol and its associated microbial metabolites in this review."
A synthetic presentation in a table of food sources of ellagitannins and isoflavones with their content specification and with the enumeration of the most important structures of these structural categories could be added in the chapter 2.
We thank the reviewer for this suggestion. We included two tables, Table 1 (p. 4-5) and Table 3 (p. 9), summarizing the content of these polyphenols in foods.
All of this could ensure a better understanding of the text and an improvement in the value of the manuscript.
We thank the reviewer for their comments and we believe that the manuscript benefited from them.
Reviewer 4 Report
The purpose of the presented manuscript was to systematize the information from the specialized literature regarding the possible contribution of natural polyphenols metabolites to cardiovascular protection.
The manuscript is interesting and useful due to the very current topic addressed, for new research in the field.
The polyphenols represent a large class of natural compounds, which can contribute to human health, through medicinal or dietary intake. One of the biggest problems that appears in the clinical utilization of these compounds is their poor bioavailability and more and more studies today are oriented in this direction. The different response of some patients to the action of medicinal substances seems to be also associated with the gut microbiota of each individual, which can influence the formation of active metabolites. In this context, the authors discussed the influence of the gut microbiota on the metabolism of some polyphenols derived from ellagic acid and some phytoestrogens respectively, which through their antioxidant potential, can have a cardioprotective effect. The authors used a large number of scientific articles from which they extracted valuable information. The results of previous researches are well presented, in a clear manner, with a good interpretation of the data and appropriate comments.
Author Response
The purpose of the presented manuscript was to systematize the information from the specialized literature regarding the possible contribution of natural polyphenols metabolites to cardiovascular protection.
The manuscript is interesting and useful due to the very current topic addressed, for new research in the field.
The polyphenols represent a large class of natural compounds, which can contribute to human health, through medicinal or dietary intake. One of the biggest problems that appears in the clinical utilization of these compounds is their poor bioavailability and more and more studies today are oriented in this direction. The different response of some patients to the action of medicinal substances seems to be also associated with the gut microbiota of each individual, which can influence the formation of active metabolites. In this context, the authors discussed the influence of the gut microbiota on the metabolism of some polyphenols derived from ellagic acid and some phytoestrogens respectively, which through their antioxidant potential, can have a cardioprotective effect. The authors used a large number of scientific articles from which they extracted valuable information. The results of previous researches are well presented, in a clear manner, with a good interpretation of the data and appropriate comments.
We thank the reviewer for taking the time to review our manuscript and for their careful consideration.
Reviewer 5 Report
The manuscript presents an interesting review, devoted to an actual problem of the effects of polyphenol-derived microbiota metabolites on cardiovascular health in humans. The review is not a systematic review and criteria of selection of source data are not described; apparently most of the available, not so abundant, relevant literature data have been considered. The review is based on over 100 literature references. The Conclusions are scientifically sound.
Remarks:
The question of the bioavailability of urolithins could be discussed including the presentation of relevant data if available.
Are all data reported in Table 1 specific for ellagitannins? Pomegranates and berries are often employed in studies of the effects of anthocyanins.
Please specify the criteria for the search and inclusion of data.
Figure 1. The terms should refer consequently to processes (e.g. absorption) avoiding expressions such as “reabsorb and excrete”
Data on the bioavailability of polyphenolsand their metabolites could be presented (if available)
Are all data reported in Table 1 specific for ellagitannins? Pomegranates and berries are often employed in studies of the effects of anthocyanins.
Author Response
The manuscript presents an interesting review, devoted to an actual problem of the effects of polyphenol-derived microbiota metabolites on cardiovascular health in humans. The review is not a systematic review and criteria of selection of source data are not described; apparently most of the available, not so abundant, relevant literature data have been considered. The review is based on over 100 literature references. The Conclusions are scientifically sound.
We thank the reviewer for taking the time to review our manuscript and for their careful consideration.
Remarks:
The question of the bioavailability of urolithins could be discussed including the presentation of relevant data if available.
Please see our response below.
Are all data reported in Table 1 specific for ellagitannins? Pomegranates and berries are often employed in studies of the effects of anthocyanins.
We thank the reviewer for this comment. Yes, we confirm that all data reported in Table 1 (Table 2 in the revised manuscript) are specific for ellagitannins.
Detail comments
Please specify the criteria for the search and inclusion of data.
We thank the reviewer for this comment. We detailed the criteria for identifying studies for the review in the revised paper (p. 4, lines 156-161), as follows:
"For this review, we searched PubMed utilizing the search strings “(urolithins OR ellagic acid OR ellagitannins) AND (cardiovascular health OR cardiovascular disease)” and “(equols OR isoflavones) AND (cardiovascular health OR cardiovascular disease)” to identify studies. The retrieved studies were screened using the following inclusion criteria: (i) observational or interventional human studies, (ii) main outcome related to cardiovascular health, (iii) published in English, and (iv) published in the last 10 years."
Figure 1. The terms should refer consequently to processes (e.g. absorption) avoiding expressions such as “reabsorb and excrete”
We thank the reviewer for drawing our attention to this inconsistency. We have corrected Fig. 1 accordingly.
Data on the bioavailability of polyphenolsand their metabolites could be presented (if available)
We thank the reviewer for this suggestion. We included information on the bioavailability of urolithins (lines 176-178) and equols (lines 274-277).
Are all data reported in Table 1 specific for ellagitannins? Pomegranates and berries are often employed in studies of the effects of anthocyanins.
As mentioned above, we confirm that all data reported in Table 1 (Table 2 in the revised manuscript) are specific for ellagitannins.